# CanonNet: Spectral Canonicalization and Curvature-Driven Learning for Compact Local-Geometry Point-Cloud Operators

## Abstract

To address the persistent challenges of scalability and robust local geometry representation in point-cloud processing, we propose **CanonNet**, a highly efficient local feature operator. CanonNet first employs **spectral canonicalization** to establish an *invariant local frame* for each neighborhood. It then uses a **geometric learning framework**, trained on synthetic surfaces, to distill *fundamental curvature priors* into a lightweight MLP. This design allows CanonNet to achieve competitive performance on various benchmarks with a $100\times$ reduction in parameters, while also exhibiting robust domain transfer. Its efficiency and design make it an effective building block for deep, hierarchical models, acting as a geometric analogue to the convolution operator for capturing multi-scale features.

## 1 Introduction

Point clouds, which are unstructured collections of 3D points, have become fundamental to numerous applications including autonomous driving (Li et al., 2020b), robotics (Pomerleau et al., 2015), and medical imaging (Sitek et al., 2006). While point clouds capture detailed geometric information, their unstructured nature presents significant challenges for processing and analysis. Existing approaches have not fully resolved two critical challenges (1) establishing a consistent ordering of points and (2) effectively learning fine-grained geometric features. In this paper, we present *CanonNet*, a novel approach that establishes canonical point ordering and orientation while enhancing the learning of geometric features through synthetic data with *precise* curvature annotations.

The unstructured nature of point clouds necessitates neural architectures that are permutation-invariant (Qi et al., 2017a;b; Guo et al., 2021; Pan et al., 2021; Wang et al., 2019). This is typically achieved through operations like pooling, which aggregate point features regardless of their order. PointNet Qi et al. (2017a) pioneered the application of such operations in point cloud processing (*e.g.* classification, segmentation ), though similar permutation-invariant mechanisms had already been explored in Graph Neural Networks (GNNs) (Behler & Parrinello, 2007; Duvenaud et al., 2015). While effective in ensuring order invariance, these symmetric aggregation functions inherently limit a network's expressivity (Kondor et al., 2018; de Haan et al., 2020), restricting its ability to capture fine-grained geometric relationships (Joshi et al., 2023).

To enhance neural network expressivity when processing graphs, researchers have developed various positional encoding (PE) methods (Grötschla et al., 2024). such as Laplacian-based, random walk-based, and other approaches. These techniques, particularly those using Laplacian eigenvectors, effectively encode structural properties (Belkin & Niyogi, 2003; Kreuzer et al., 2021; Dwivedi et al., 2023; Maskey et al., 2022). However, in point cloud processing, PE has primarily been limited to Transformer-based architectures (Lai et al., 2022; Zhao et al., 2021; Qin et al., 2022; Pan et al., 2021), with some Transformer variants deliberately omitting it (Guo et al., 2021). We show that Laplacian PE can be harnessed in point cloud pro-

cessing to achieve canonical order and orientation, eliminating the need for complex transformation-invariant architectures

The geometric properties inherent in point clouds constitute another valuable source of information that can be harnessed by neural architectures. Rather than relying solely on learned representations, various approaches exploit explicitly computed features such as normals (Deng et al., 2018a;b; Yuan et al., 2023), angles (Deng et al., 2018a;b; Yuan et al., 2023; Qin et al., 2022), and pairwise distances (Deng et al., 2018a;b; Yuan et al., 2023; Qin et al., 2022) as supplementary inputs to enhance model capabilities. Among these geometric property approaches, approximating curvature directly from point clouds via triangulation and supplying them as features to neural networks has achieved significant performance gains (Ran et al., 2022). In this context, a primary constraint is the limited availability of high-quality training data with accurate geometric annotations, which has restricted the evolution of learning-based approaches that can effectively utilize these geometric properties. Real-world point cloud datasets often lack precise geometric ground truth, making it difficult to train models that can reliably learn and interpret local surface properties. This limitation has particularly affected the development of approaches that aim to understand fine-grained geometric features.

## KEY CONTRIBUTIONS

1. **Canonical Preprocessing**: A novel pipeline that establishes both *canonical ordering* and *orientation* for local point patches, ensuring invariance to point permutations and rigid transformations.

2. **Synthetic Data Generation**: A framework leveraging analytic surfaces with known curvatures, enabling *unlimited training samples* with precise geometric properties.

3. **Lightweight Architecture**: A neural network that effectively learns local geometric features through curvature-based classification.

## 2 RELATED WORKS

Point cloud processing is challenging due to the data's irregularity and lack of inherent order. Our work addresses these issues with two innovations: geometry-aware canonical ordering and curvature-based synthetic data generation. We review related literature across three areas.

**Deep Learning for Point Cloud Processing.** Early approaches converted point clouds into voxels or projections, losing geometric detail. PointNet Qi et al. (2017a) introduced direct point-based processing, extended by PointNet++ (Qi et al., 2017b) for hierarchical feature learning. DGCNN Wang et al. (2019) improved local modeling through dynamic graphs, while KPConv Thomas et al. (2019) introduced geometry-adaptive convolutions. Transformer-based models like Point Transformer Zhao et al. (2021) and PCT Guo et al. (2021) leveraged self-attention for geometric relations. Despite progress, these methods are computationally heavy and limited in capturing intrinsic geometry.

**Surface Geometry in Point Clouds.** Surface properties such as normals and curvatures are crucial for tasks like registration and classification. PCPNet Guerrero et al. (2018) learned local geometry from patches, while DeepFit Ben-Shabat & Gould (2020) applied neural surface fitting for accurate normals and curvature estimation. Incorporating geometric cues (distances, angles, normals, curvature) has consistently improved downstream performance (Rusu et al., 2008; 2009; Deng et al., 2018b; Yuan et al., 2023; Qin et al., 2022; Ran et al., 2022). Our method builds on these works, introducing an efficient curvature-based approach that avoids heavy architectures by enforcing invariances in preprocessing.

**Ordering and Orientation in Point Clouds.** Canonical alignment is another strategy for invariance. STN (Jaderberg et al., 2015) and its PointNet extension (Qi et al., 2017a) used learned transformations, while PCPNet (Guerrero et al., 2018) stabilized orientation via rotation-only constraints. Ordering meth-

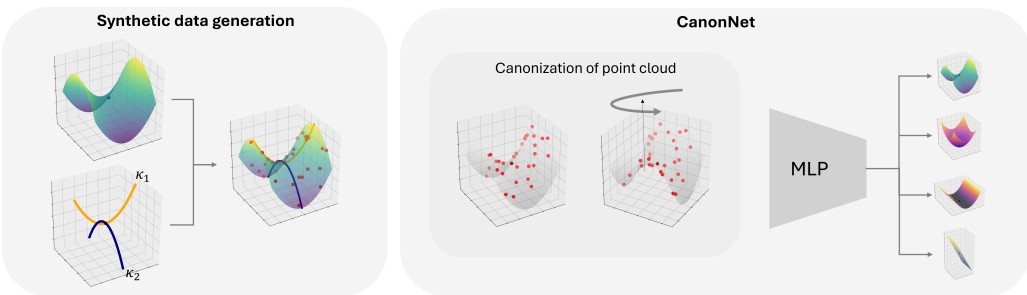

Figure 1. The complete CanonNet pipeline: Synthetic data generation (LHS): We sample points from analytically defined surfaces with known principal curvatures $\kappa_1, \kappa_2$. Processing and classification (RHS): The point cloud is transformed into canonical orientation and processed by an MLP that performs supervised classification into four geometric surface types (Saddle, Parabolic, Valley, Plane) using ground truth curvature labels.

ods (Zheng et al., 2019; Yang et al., 2023; Dovrat et al., 2019) often focus on downsampling rather than true canonical ordering. In contrast, our approach unifies canonical ordering and orientation with curvature-based synthetic supervision, enabling lightweight, permutation-invariant processing with strong geometric consistency.

## 3 METHOD

We present a framework that combines differential geometry with deep learning to enable efficient point cloud processing. Our approach consists of two main components: (1) a preprocessing pipeline that establishes canonical point ordering and orientation, and (2) a training methodology utilizing synthetically generated surfaces with known geometric properties.

### 3.1 PREPROCESSING PIPELINE

This section details our preprocessing approach for creating consistent point ordering and orientation.

**Local Patch Extraction.** For each query point $p_q$ in the 3D point cloud $\boldsymbol{P}$, we extract a local patch $\boldsymbol{X}$ using $k$-nearest neighbors ($k = 20$). This effectively captures the surrounding geometry of the point.

**Graph Construction and Spectral Embedding.** Given a local patch from a 3D point cloud $\boldsymbol{X} = \{x_i\}_{i=1}^N, x_i \in \mathbb{R}^3$ or in matrix form $\boldsymbol{X} \in \mathbb{R}^{N \times 3}$, we aim to establish a consistent point ordering that is invariant to permutations and rigid transformations. To achieve this, we construct a fully connected, undirected graph $\mathcal{G} = (\mathcal{V}, \boldsymbol{W})$ to capture the local geometric relationships between points. The edge weights $\boldsymbol{W}$ are defined by the heat kernel:

$$\boldsymbol{W}_{ij} = \exp\left(-\frac{\|\boldsymbol{x}_i - \boldsymbol{x}_j\|_2^2}{t}\right) \quad \forall \boldsymbol{x}_i, \boldsymbol{x}_j \in \mathcal{V} \tag{1}$$

Here $t > 0$ is a temperature parameter controlling the locality of point interactions. We compute the normalized graph Laplacian (Chung, 1997):

$$\boldsymbol{L} = \Delta^{-1/2}(\Delta - \boldsymbol{W})\Delta^{-1/2} \tag{2}$$

where $\Delta \in \mathbb{R}^{N \times N}$ is the diagonal degree matrix with entries $\Delta_{ii} = \sum_{j=1}^N \mathbf{W}_{ij}$ for $i = 1, 2, \ldots, N$, and $\Delta_{ij} = 0$ for all $i \neq j$.

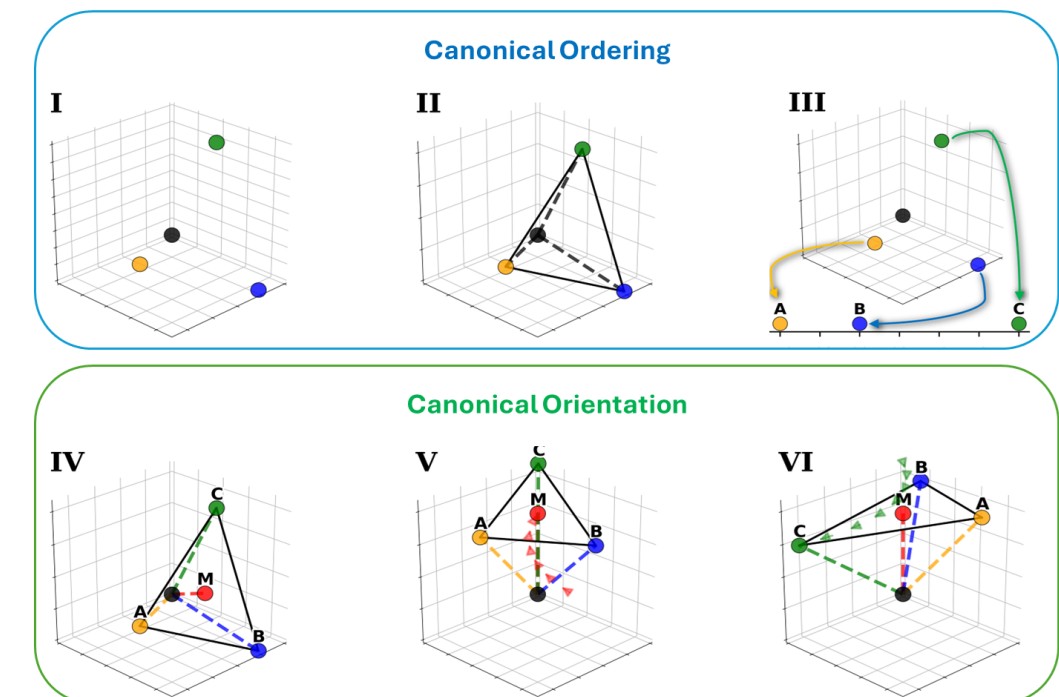

Figure 2. Illustration of the preprocessing pipeline for establishing canonical point cloud representation as described in Section 3.1: **(I)** Input point cloud with arbitrary ordering and orientation. **(II)** Construction of fully connected graph with heat kernel weights and computation of normalized graph Laplacian. **(III)** Reordering points along a 1D axis based on Laplacian eigenvector values, ensuring consistency regardless of initial point indexing or spatial orientation and position. **(IV)** Identification of geometric landmarks: center of mass, 'M', and the point corresponding to the largest eigenvector value, 'A'. **(V)** First standardization rotation aligning center of mass with positive z-axis. **(VI)** Second standardization rotation placing 'A' in the XZ-plane with positive x-coordinate, completing the transformation pipeline that ensures both permutation and rigid-transformation invariance.

Next, we compute the eigenvector $\phi \in \mathbb{R}^N$ corresponding to the smallest nonzero eigenvalue of $\mathbf{L}$ (Fiedler, 1973; 1975). This eigenvector establishes a spectral embedding where each point $\mathbf{x}_i$ in the point cloud corresponds to the $i$-th component of $\phi$, effectively projecting the 3D points onto a single line. As demonstrated by (Belkin & Niyogi, 2003), this embedding minimizes the weighted sum $\sum_{i,j} \mathbf{W}_{ij}(\phi_i - \phi_j)^2$, ensuring that points with strong connections in the original 3D space remain close in the 1D embedding, thereby optimally preserving the local geometric structure. This preservation of local geometric structure enhances robustness to noise, as reflected in the performance gains reported in Section A.4.

It is worth emphasizing that the eigenvector computation for small patches is computationally inexpensive. This efficiency can be further optimized through established iterative methods such as the power method or Lanczos algorithm, preserving the lightweight nature of our preprocessing approach.

**Canonical Ordering and Orientation.** As illustrated in Fig. 2, we establish a standardized point cloud representation through three complementary transformations that address ordering, position, and orientation. Together, these transformations ensure that geometrically equivalent shapes converge to identical representations regardless of their initial configurations.

The spectral embedding from the previous step forms the basis of our canonical ordering. Sorting the points by their values in the eigenvector $\phi$ defines a permutation $\sigma \in S_N$, such that $\phi_{\sigma(1)} \leq \phi_{\sigma(2)} \leq \cdots \leq \phi_{\sigma(N)}$.

We construct the corresponding permutation matrix $\mathbf{\Pi}$, where $\Pi_{ij} = 1$ if $j = \sigma(i)$ and 0 otherwise. This matrix is applied to the point cloud, yielding the reordered set of points as follows:

$$\bar{\mathcal{X}} = \mathbf{\Pi}\mathcal{X} \tag{3}$$

Next, we normalize the positions by aligning the center of mass with the z-axis. From the reordered point cloud, we compute the center of mass as follows:

$$m = \frac{1}{N} \sum_{i=1}^{N} \bar{\boldsymbol{x}}_i \tag{4}$$

Next, we compute a rotation matrix $\mathbf{R}_{cm}$ that places $m$ at $(0, 0, \|m\|_2^2)$. Applying this rotation results in:

$$\mathcal{Y} = \mathbf{R}_{cm}\bar{\mathcal{X}} \tag{5}$$

To ensure consistent orientation, we apply a final rotation about the z-axis based on a landmark point. Specifically, we identify the point $p_1 = (x_1, y_1, z_1)$ in $\mathcal{Y}$ that corresponds to the largest value in $\phi$ and compute a rotation $\mathbf{R}_z$ that places $p_1$ into the $XZ$-plane with a positive x-coordinate. This yields the final standardized representation:

$$\mathcal{P} = \mathbf{R}_z\mathcal{Y} \tag{6}$$

The full transformation pipeline is as follows:

$$\mathcal{P} = \mathbf{R}_z\mathbf{R}_{cm}\mathbf{\Pi}\mathcal{X} \tag{7}$$

**Canonical Ordering and Orientation.** A key advantage of our preprocessing pipeline is its theoretical guarantees of invariance to both point permutations and rigid transformations. We formally establish these properties below.

**Theorem 1.** *The proposed preprocessing pipeline is invariant to point permutation and rigid transformation.*

*Proof.* Let $\boldsymbol{X} \in \mathbb{R}^{N \times 3}$ be a 3D point cloud, $\boldsymbol{P} \in \mathbb{R}^{N \times N}$ a permutation matrix, $\boldsymbol{R} \in \mathbb{R}^{3 \times 3}$ a rotation matrix, and $\boldsymbol{t} \in \mathbb{R}^3$ a translation vector. We define the transformed point cloud as $\tilde{\boldsymbol{X}} = \boldsymbol{R}\boldsymbol{X} + \boldsymbol{t}$.

**Rigid Transformation Invariance:** For any pair of points $\boldsymbol{x}_i, \boldsymbol{x}_j \in \boldsymbol{X}$, their pairwise distance remains unchanged under rigid transformation:

$$\|\tilde{\boldsymbol{x}}_i - \tilde{\boldsymbol{x}}_j\| = \|(\boldsymbol{R}\boldsymbol{x}_i + \boldsymbol{t}) - (\boldsymbol{R}\boldsymbol{x}_j + \boldsymbol{t})\| = \|\boldsymbol{R}(\boldsymbol{x}_i - \boldsymbol{x}_j)\| = \|\boldsymbol{x}_i - \boldsymbol{x}_j\| \tag{8}$$

Thus, the weight matrix $\boldsymbol{W}$ remains invariant, leading to an identical Laplacian matrix. **Point Permutation Invariance:** Let $\phi$ be the eigenvector corresponding to the smallest non-zero eigenvalue $\lambda$ of the Laplacian matrix $\boldsymbol{L}$. We assume that the multiplicity of $\lambda$ is 1 (see A.3), ensuring that $\phi$ is unique up to scaling. For a permuted point cloud $\bar{\boldsymbol{X}} = \boldsymbol{P}\boldsymbol{X}$, with permutation matrix $\boldsymbol{P}$, the weight matrix transforms as $\bar{\boldsymbol{W}} = \boldsymbol{P}\boldsymbol{W}\boldsymbol{P}^{-1}$, resulting in a similarity-transformed Laplacian $\bar{\boldsymbol{L}} = \boldsymbol{P}\boldsymbol{L}\boldsymbol{P}^{-1}$. For the eigenvector $\phi$ of $\boldsymbol{L}$ with eigenvalue $\lambda$, we have:

$$\bar{\boldsymbol{L}}(\boldsymbol{P}\phi) = \boldsymbol{P}(\boldsymbol{L}\phi) = \boldsymbol{P}(\lambda\phi) = \lambda(\boldsymbol{P}\phi) \tag{9}$$

Therefore, $(\boldsymbol{P}\phi)$ is an eigenvector of $\bar{\boldsymbol{L}}$ with eigenvalue $\lambda$. Since $\lambda$ has multiplicity 1, eigenvectors for the original and permuted Laplacians differ only by the permutation $\boldsymbol{P}$ and scaling.

To establish a canonical ordering, we first normalize $\phi$ so its largest-magnitude entry is positive, resolving sign ambiguity. We then permute the points according to these normalized eigenvector values, yielding a point ordering invariant to initial permutations. $\square$

These invariance properties enable our lightweight MLP architecture to focus exclusively on learning geometric features, without needing complex structures to handle permutation and transformation equivariance.

These invariance properties ensure that our preprocessing pipeline produces consistent results regardless of the initial point ordering or orientation of the input point cloud. This theoretical guarantee enables our lightweight MLP architecture to focus on learning the geometric properties of the surface rather than accounting for permutation and transformation variations.

### 3.2 TRAINING

This section outlines our geometric learning framework. We first present our synthetic data generation approach with controlled curvature properties. Then we describe our geometric feature extraction process and the design of our parameter-efficient network for surface classification.

**Synthetic Data Generation.** Surface curvature quantifies how a surface deviates from being flat at a point. The principal curvatures $\kappa_1$ and $\kappa_2$ represent the maximum and minimum bending of the surface. From these, we derive the Gaussian curvature $K = \kappa_1 \kappa_2$ and the mean curvature $H = \kappa_1 + \kappa_2$. For our dataset, we generate quadratic surfaces with analytically tractable curvature properties:

$$z = f(x, y) = ax^2 + by^2 + cxy + dx + ey \tag{10}$$

Among possible sampling methods, we chose to sample coefficients $(a, b, c, d, e)$, which proved sufficient to create surfaces with the full range of desired curvature characteristics. These surfaces are classified into one of four categories: $\mathcal{C} = \{\text{plane}, \text{parabolic}, \text{valley}, \text{saddle}\}$ based on its curvature signature at the origin.

To ensure an unbiased dataset, we sample each class separately with equal representation. For each generated surface, we uniformly sample points within the region $[-0.5, 0.5] \times [-0.5, 0.5]$ to create our synthetic dataset.

**Feature Extraction and Network Design.** We train a lightweight Multi-Layer Perceptron (MLP) in a supervised manner on synthetic point cloud data (see previous section). The input consists of the preprocessed point cloud $\mathcal{P}$ augmented with second-order polynomial features of each point's coordinates (e.g., $x^2, y^2, xy$). These terms capture local curvature variations and encode higher-order geometric relationships, allowing the network to better distinguish surface classes with different curvature characteristics. The MLP is trained to classify each $\mathcal{P}$ into one of four fundamental surface categories:

1. **Plane:** A flat surface characterized by zero Gaussian and mean curvatures.

2. **Parabolic:** A convex surface with positive Gaussian curvature (e.g., spheres, domes).

3. **Valley:** A surface with zero Gaussian curvature and nonzero mean curvature (e.g., a cylinder).

4. **Saddle:** A hyperbolic surface with negative Gaussian curvature.

These four surface types represent fundamental geometric structures commonly encountered in real-world point cloud data. Their classification is critical for downstream tasks such as shape reconstruction and geometric reasoning. Fig. 3 provides visual illustrations of these surfaces, highlighting their distinct curvature properties.

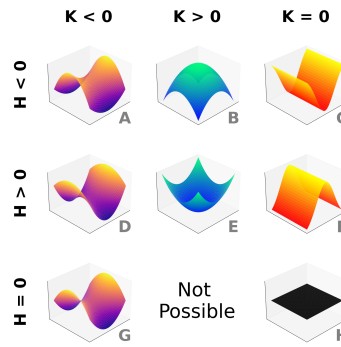

Figure 3. The different possible surfaces given the Gaussian (**K**) and mean (**H**) curvature as described in Section 3.2. Note that up to rigid motion there are 4 different types of surfaces.

## 4 EXPERIMENTS

We evaluate CanonNet on three tasks that capture essential aspects of local geometric understanding: ***curvature estimation***, ***Descriptor retrieval***, and ***surface classification***. While downstream tasks like classification or segmentation are important, these chosen experiments provide a more direct and rigorous assessment of an operator's ability to capture fine-grained surface properties. We conduct evaluations across three datasets and simulate real-world corruptions, including **partial overlap**, **random rotations**, and **Gaussian noise**, to assess the model's resilience. These experiments address the core challenge of understanding local 3D geometry, a prerequisite for a multitude of tasks from point cloud registration to surface analysis. Excelling at these geometric benchmarks validates CanonNet's effectiveness as a foundational operator for robust, real-world 3D perception.

### 4.1 GAUSSIAN AND MEAN CURVATURE ESTIMATION

**Experimental Setup.** We evaluate CanonNet on the PCPNet dataset (Guerrero et al., 2018), which includes point clouds sampled from various 3D shapes with ground truth normals and curvatures. Unlike PCPNet and DeepFit, which are trained directly on this dataset, CanonNet is trained solely on synthetic quadratic surfaces representing local geometry (Section 3.2). Our model is a lightweight MLP with only 0.03M parameters and operates on local patches of just 20 points, ordered canonically. This compactness is enabled by our canonical preprocessing and local geometric learning pipeline, which act as strong priors—allowing small models to learn powerful representations that generalize across diverse point cloud data.

**Curvature Estimation Task.** We estimate Gaussian curvature ($K$) and absolute mean curvature ($|H|$), which are invariant to normal orientation and better aligned with surface-based reasoning. CanonNet jointly predicts these curvatures along with the surface type (4 categories), encouraging deeper geometric understanding and aiding learning through multi-task supervision.

**Comparison Baselines.** We compare against PCPNet (Guerrero et al., 2018) and DeepFit (Ben-Shabat & Gould, 2020), two baselines based on PointNet with permutation-invariant pooling. PCPNet uses 500-point patches and 22M parameters; DeepFit uses 128-point patches and 3.5M parameters. In contrast, CanonNet uses only 20 points per patch and 0.03M parameters—roughly **700×** and **116×** smaller, respectively.

**Evaluation Metric.** We evaluate curvature estimation performance using the rectified error metric, which is calculated as:
$$D_K = \frac{|K - K_{GT}|}{\max\{|K_{GT}|, 1\}}, \quad D_H = \frac{|H - H_{GT}|}{\max\{|H_{GT}|, 1\}}, \tag{11}$$
where $K$ and $H$ are the predicted Gaussian and mean curvatures, and $K_{GT}$ and $H_{GT}$ are the ground truth values. The final error metrics are reported as the root mean square error (RMSE) of these normalized differences. This metric normalizes the error by the maximum of the absolute ground truth value and 1.0, ensuring stable evaluation across regions with different curvature magnitudes.

**Results and Analysis.** Section 4.3 presents the quantitative results. CanonNet achieves a mean curvature error of 0.40 on the PCPNet dataset—surpassing both PCPNet (1.91) and DeepFit (0.67)—despite being trained on synthetic data and using dramatically smaller patches. Its Gaussian curvature error is higher (8.2) but remains competitive considering the scale of the model and input.

On synthetic surfaces with analytical curvature, CanonNet achieves strong accuracy: 0.97 (Gaussian) and 0.14 (mean). These results demonstrate that our preprocessing and geometric learning pipeline effectively substitute for architectural complexity, enabling compact models to match or exceed the performance of much larger networks. These methods, though foundational, are widely used as benchmarks for geometric feature learning because of their strong performance and the scarcity of datasets with reliable ground-truth annotations.

## 4.2 GEOMETRIC DESCRIPTOR RETRIEVAL

**Experimental Setup.** Unlike other methods explicitly trained for descriptor matching, CanonNet is not trained as a geometric descriptor. Instead, we leverage its learned geometric understanding from curvature estimation and surface classification. Descriptors are formed by applying CanonNet to multi-resolution patches (via progressive downsampling) and concatenating the resulting features.

We evaluate CanonNet on the 3DMatch benchmark (Zeng et al., 2017) using the Feature Match Recall (FMR) (Deng et al., 2018a) metric, which measures the fraction of ground-truth correspondences correctly matched via mutual nearest neighbors in descriptor space. As in prior work, the benchmark uses point cloud pairs with at least 30% overlap. We test models trained and evaluated on the same dataset (*In-Domain*) and those trained on one and tested on another (*Cross-Domain*). Notably, CanonNet is trained exclusively on synthetic surface data capturing only local geometric properties.

**Results and Analysis.** Section 4.3 compares CanonNet to both traditional (FPFH (Rusu et al., 2009), SHOT (Tombari et al., 2010)) and learned descriptors (3DMatch (Zeng et al., 2017), CGF (Khoury et al., 2017), PerfectMatch (Gojcic et al., 2019), FCGF (Choy et al., 2019), D3Feat (Bai et al., 2020), LMVD (Li et al., 2020a), SpinNet (Ao et al., 2021)). While some of these methods were developed several years ago, they remain strong and relevant benchmarks in the field (Jung et al., 2024).

CanonNet achieves a competitive 65.7% FMR on unseen data, despite never being trained for this task. Spin-Net reaches 92.8% but uses $21.6\times$ more parameters; LMVD scores 79.9% with $26.6\times$ more. CanonNet is by far the smallest model: just 0.03Mb in size. This compactness results from a combination of our canonical preprocessing, which removes the need for complex architectures to handle point permutations or rigid transformations, and a geometric learning pipeline that effectively captures local surface structure—applicable to arbitrary point clouds.

Efficiency extends beyond model size. CanonNet processes only 300 points per patch (across resolutions), compared to 1000–2000 for others. Despite a lower inlier rate, it needs just 5 RANSAC iterations on average to find valid correspondences—offset by a $30\times$ speedup in descriptor generation over SpinNet on standard hardware. This makes CanonNet particularly well-suited for real-time or resource-constrained applications.

## 4.3 SURFACE CLASSIFICATION

**Experimental Setup.** To further test CanonNet's robustness, we evaluate it on a synthetic surface classification task under randomized and noisy conditions. At test time, each surface instance is randomly sampled from one of several unseen surface shapes, then independently subjected to random point permutations, arbitrary 3D rotations, and additive Gaussian noise with varying magnitudes (0%–10%). This setup challenges the model's ability to maintain accurate classification under conditions of structural variation, noise, and transformation—mimicking real-world deployment scenarios.

**Results and Analysis.** Fig. 5 shows the classification accuracy across noise levels. CanonNet achieves high accuracy on clean data (98%), and maintains strong performance under moderate noise—retaining 80% accuracy even at 10% noise. This demonstrates the model's effectiveness in extracting stable local geometry from deformed data, validating our canonical preprocessing and learning design.

| PCPNET dataset | | | | |
|---|---|---|---|---|
| Method | $D_K \downarrow$ | $D_H \downarrow$ | #Params (M) | #Points |
| PCPNET (Guerrero et al., 2018) | 6.88 | 1.91 | 22 | 500 |
| DeepFit (Ben-Shabat & Gould, 2020) | 0.56 | 0.67 | 3.5 | 128 |
| **CanonNet** | 8.2 | **0.4** | **0.03** | **20** |
| Synthetic dataset | | | | |
| **CanonNet** | 0.97 | **0.14** | **0.03** | **20** |

Table 1. Curvature errors and efficiency (Section 4.1). CanonNet is small, with SOTA results.

| Method | Param. (Mb) | In-Domain (%) | Cross-Domain (%) |
|---|---|---|---|
| FPFH (Choy et al., 2019) | - | 35.9 | 22.1 |
| SHOT (Tombari et al., 2010) | - | 23.8 | 61.1 |
| 3DMatch (Zeng et al., 2017) | 13.40 | 59.6 | 16.9 |
| CGF (Khoury et al., 2017) | 1.86 | 58.2 | 20.2 |
| PerfectMatch (Gojcic et al., 2019) | 3.26 | 94.7 | 79.0 |
| FCGF (Choy et al., 2019) | 33.48 | 95.2 | 16.1 |
| D3Feat (rand) (Bai et al., 2020) | 13.42 | 95.3 | 26.2 |
| LMVD (Li et al., 2020a) | 2.66 | 97.5 | 79.9 |
| SpinNet (Ao et al., 2021) | 2.16 | 97.6 | 92.8 |
| **CanonNet** | **0.03** | - | 65.7 |

Table 2. Parameter count and FMR (Section 4.2). CanonNet is compact yet competitive

## 5 CONCLUSION

We introduced CanonNet, a lightweight point cloud operator that achieves geometric invariance through a novel spectral canonicalization pipeline and learns curvature priors from synthetic data. Our approach achieves state-of-the-art mean curvature estimation with a parameter footprint orders of magnitude smaller (0.03M) than existing methods. CanonNet's principled design and high efficiency establish a new foundation for scalable and robust geometric learning, particularly in resource-constrained environments.

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

## A  Technical Appendices and Supplementary Material

### A.1  Reproducibility: Computational Resources and Training Configuration

All experiments were conducted on a single NVIDIA T4 GPU with 16GB of memory. The model used was a Multi-Layer Perceptron (MLP) with 5 layers and 64 neurons per layer, totaling approximately 30,000 parameters. Training was performed using the Adam optimizer with an initial learning rate of 0.1, decaying every 30 epochs. The training dataset consisted of 50,000 samples, with 5,000 samples used for testing. The model was trained for 100 epochs, requiring approximately 2 hours of execution time. This setup is reproducible on mid-range GPU hardware with similar compute capabilities.

### A.2  Scale Invariance

Our model is designed to be invariant to scale. This is achieved through a normalization method applied to the input point cloud. Specifically, we scale the entire point cloud such that the median size of its local patches is equivalent to a unit cube. This normalization ensures that all local operators within the model process patches at a consistent scale, regardless of the original scale of the input data.

This approach is crucial because while the absolute distances between points are altered by this scaling, the **relative distances** and spatial relationships between points within the point cloud remain unchanged. This preserves the essential geometric structure of the data, allowing the model to generalize effectively across inputs of varying sizes without being affected by the differences in their overall scale.

### A.3  Uniqueness of the Fiedler Value

The Fiedler value, or the second smallest eigenvalue of the graph Laplacian, is theoretically unique only if its multiplicity is one. While perfect symmetry in a graph can lead to a degenerate Fiedler value (i.e., a multiplicity greater than one), this scenario has minimal practical impact on our method due to the following considerations:

1. **Real-World Data:** Our method is applied to real-world data, which is inherently noisy. Infinitesimal perturbations from noise or sampling variations are ubiquitous and sufficient to break perfect symmetry within a local patch, thereby ensuring the Fiedler value's multiplicity is one.

2. **Local Scope:** Our algorithm operates on small, local patches rather than large, globally symmetric objects. The probability of a randomly sampled local patch exhibiting perfect symmetry is negligible, even when extracted from a globally symmetric shape.

3. **Graceful Degradation:** In the exceedingly rare event that the Fiedler value is degenerate, our method does not fail. Any eigenvector from the degenerate subspace can serve as a valid, albeit potentially less stable, canonicalization of the local patch.

For these reasons, the assumption that the Fiedler value is unique is valid for practical applications, and its use in spectral graph theory, particularly in graph partitioning, is well-established.

### A.4  Ablation Studies

We performed an ablation study to systematically evaluate the contribution of four key components in the CanonNet architecture: graph Laplacian formulations, the preprocessing pipeline, second-degree polynomial features, and Laplacian eigenvalues as supplementary input features.

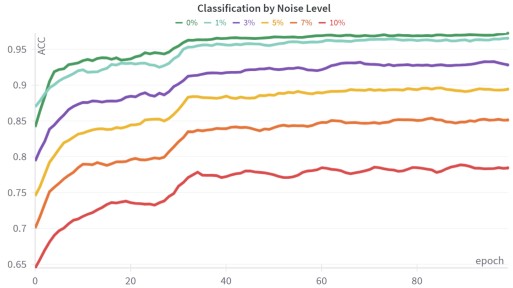

Figure 4. **(a)** Impact of canonical preprocessing pipeline, showing consistent 10-17% accuracy improvements across architectures and noise levels (solid: baseline, dashed: with preprocessing). **(b)** Effect of second-degree polynomial features, yielding approximately 5% accuracy improvement across all tested architectures (solid: baseline, dashed: with polynomial features). **(c)** Impact of Laplacian eigenvalues, showing minimal differences (±0.2%), suggesting geometric information is already well-captured by existing features.

Figure 5. Noise effect on surface classification.

| Temp. / Noise | 0 | 1% | 3% | 5% | 7% | 10% |
|---|---|---|---|---|---|---|
| t=0.5 (Norm) | 100 | 83 | 62.76 | 49.86 | 41.33 | 33.19 |
| t=0.5 | 100 | 83.19 | 63 | 50.90 | 43.19 | 34.76 |
| t=1 (Norm) | 100 | 83 | 63.38 | 51.05 | 42.57 | 34.57 |
| t=1 | 100 | 82.05 | 62.19 | 50.67 | 42.38 | 34.24 |
| t=2 (Norm) | 100 | 83.48 | 63.19 | 51 | 43.43 | 34.76 |
| t=2 | 100 | 81.90 | 61.38 | 49.86 | 42.24 | 33.48 |
| t=5 (Norm) | 100 | 83.19 | 62.81 | 51.33 | 43.43 | 34.71 |
| t=5 | 100 | 81.95 | 61.76 | 49.67 | 41.29 | 33.48 |

Table 6. Laplacian normalization and temperature effects on point ordering (Section A.4).

### Graph Laplacian Selection.

Results in Section A.4, show normalized graph Laplacians generally perform slightly better than unnormalized versions when exposed to noise. This advantage remains consistent across all noise levels tested. Since different temperature settings produced nearly identical results, we selected $t = 1$ for our normalized formulation implementation.

**Impact of Canonical Preprocessing Pipeline.** Our canonical preprocessing pipeline boosted classification across all models. Fig. 4 (a) shows 10% accuracy gains from using canonical ordering and orientation, confirming its role in addressing permutation and rotation invariance. Under Gaussian noise, the gap widened to nearly 15%, emphasizing its contribution to robust geometric learning.

**Second-Degree Polynomial Features.** Adding second-degree polynomial terms improved accuracy by 5% across all models, as shown in Fig. 4 (b), with deeper networks benefiting more. These results support our hypothesis that such terms enhance curvature modeling through geometry-aligned features, enabling even simple networks to better distinguish surface types.

**Laplacian Eigenvalues as Input Features.** Incorporating Laplacian eigenvalues, motivated by their link to intrinsic geometry, had negligible impact (±0.2% in Fig. 4 (c)). This suggests our existing inputs (coordinates + polynomial terms) already capture the necessary geometric structure.