# OpenReview forum: "CanonNet: Spectral Canonicalization and Curvature-Driven Learning for Compact Local-Geometry Point-Cloud Operators"
_ICLR.cc/2026/Conference — Submitted to ICLR 2026_

### Official Review · Reviewer_8aXT · 2025-10-27

**Soundness:** 2
**Presentation:** 2
**Contribution:** 2
**Rating:** 4
**Confidence:** 3

**Summary:**

The paper proposes CanonNet, which attempts to extract local geometric feature. It trains on synthetic surfaces to distill fundamental curvature priors to a light-weight MLP. It shows reasonable performance on curvature prediction.

**Strengths:**

The idea of canonical order and orientation seems useful in local point cloud feature extraction. The idea of distilling curvature priors to MLP is interesting.

**Weaknesses:**

* The paper lacks a justification behind the proposed method. For example, why the canonical order and orientation is defined in such way. Is it robust to any transform?
* The synthetic curvature is too simple and cannot capture real-world complex point clouds.
* The experiments evaluation is weak. It only consists of two synthetic dataset without any real-world tasks, such as classification and segmentation.
* After all above issues are addressed, the writing needs to be improved substantially.

**Questions:**

No.

---

> ### Author Response · Authors · 2025-11-18
>
> We thank the reviewer for their time and feedback. We are encouraged that the reviewer recognized the utility and interest in our core ideas of canonicalization and curvature-distillation.
>
> We appreciate the opportunity to clarify some key points, particularly a significant misunderstanding regarding our experimental validation, and to provide the justifications requested. We believe addressing these points will demonstrate the soundness and contribution of our work.
>
> ---
>
> ### 1. On W3 (Experimental Evaluation and Real-World Tasks)
>
> We respectfully wish to correct a significant misunderstanding regarding the experimental evaluation. Our evaluation is primarily conducted on **two standard real-world benchmarks**, not synthetic ones, as suggested.
>
> * **3DMatch (Section 4.2):** This is a large-scale, real-world benchmark for geometric descriptor matching, using scans of indoor scenes.
> * **PCPNet Dataset (Section 4.1):** As stated in Sec. 4.1, this dataset includes point clouds sampled from various 3D shapes. It is a standard benchmark for evaluating curvature estimation on real-world scanned objects.
>
> Our paper focuses on validating CanonNet as a foundational **local geometric operator**.  We therefore selected tasks that provide a more direct and rigorous assessment of an operator's ability to capture fine-grained surface properties (i.e., curvature and descriptors). We believe succeeding at these fundamental, real-world tasks is the most direct way to validate our contribution.
>
> ---
>
> ### 2. On W2 (Simplicity of Synthetic Data)
>
> We thank the reviewer for this point, as it highlights a key **strength** of our approach: **generalization**.
>
> Our model is **trained *only* on these simple synthetic surfaces**. The fact that it then achieves competitive (and in the case of mean curvature, state-of-the-art) performance on complex, **unseen real-world data** (PCPNet ]and 3DMatch) is the central validation of our method.
>
> This demonstrates that our lightweight MLP, guided by our canonicalization pipeline, successfully distills fundamental geometric priors from simple data, which then **generalize robustly** to complex real-world geometry. This was a deliberate design choice to prove our operator learns the underlying principles of local shape, not just overfitting to a specific dataset.
>
> ---
>
> ### 3. On W1 (Justification and Robustness of Canonicalization)
>
> We apologize if the justification for our method (W1) was not sufficiently clear. The justification and proof of invariance are provided in **Section 3.1** and formally established in **Theorem 1**.
>
> * **Justification:** We define the canonical ordering using the eigenvector of the normalized graph Laplacian (the Fiedler vector). As discussed in Sec. 3.1, this spectral embedding is a principled choice because it optimally preserves the local geometric structure" by ensuring points with strong connections (small distances) remain close in the 1D embedding.
> * **Robustness:** We formally prove that our *entire* preprocessing pipeline is **invariant to both point permutation and rigid transformations** (rotation and translation) in **Theorem 1** . This theoretical guarantee directly addresses the reviewer's question and is a cornerstone of our method, as it is precisely this invariance that enables a simple, lightweight MLP to be so effective.
>
> ---
>
> ### 4. On W4 (Writing)
>
> We thank the reviewer for this feedback. We will perform a thorough pass of the entire manuscript to improve clarity, refine the prose, and correct any typographical errors to enhance readability in the final version.
>
> ---
>
> We hope these clarifications resolve the reviewer's concerns, especially the misunderstanding about our use of real-world benchmarks. Given that our method is validated on real data (PCPNet, 3DMatch), demonstrates strong generalization from simple synthetic data, and is backed by a formal proof of invariance (Theorem 1), we respectfully believe it meets the bar for soundness and contribution. We are hopeful the reviewer will reconsider their evaluation in light of these clarifications.

---

### Official Review · Reviewer_dLxm · 2025-10-27

**Soundness:** 2
**Presentation:** 4
**Contribution:** 2
**Rating:** 4
**Confidence:** 4

**Summary:**

This paper proposes a novel approach for canonicalizing local patches of a point cloud. Specifically, given a local set of points, a local Laplacian is formed and it's Fiedler vector is computed to canonically order the points in this patch. From this ordering, a local SO(3) frame is computed and the patch is expressed in its coordinate system, providing a canonical layout of the local patch. Subsequently, the authors propose to train a simple MLP on a synthetic dateset, which makes use of the proposed canonicalization to predict the local surface curvature of a given patch.  After these two stages, the pre-trained MLP can be used downstream for a variety of tasks.

**Strengths:**

The method for frame canonicalization is clever, makes uses of tried-and-true computational tools, and could potentially have broader applications than proposed by the authors.   This paper is also very well written and easy to understand. The proposed method is also straightforward, and appears easy to apply.

**Weaknesses:**

While the method itself is straightforwards, the paper has several weaknesses as follows:

- Experiments are not compelling. Curvature estimation is a toy task and despite the authors statement, the model is not competitive with existing approaches on the descriptor retrieval task (with an almost 30% difference in accuracy in favor of prior approaches). Unfortunately, parameter efficiency is not very relevant. A more compelling version of this experiment would investigate the properties of the proposed MLP as a representation learner, where the pre-trained MLP is frozen and a second network trained on the features to predict descriptors in a supervised manner.

- The frame canonicalization is interesting and relevant (though I have several outstanding questions, see below), and could be more broadly applied than what is considered by the authors here (e.g. as a replacement for estimated frames in 3D equivariant networks, etc.). However, tying this approach to a simple MLP and somewhat unconventional and empirically questionable pre-training regime does not seem to be the best application of canonicalization.  Specifically, the paper does not provide compelling evidence that curvature awareness is a key component of learned descriptors, nor that it makes a useful target for representation learning.  I think this paper would be more compelling if the authors instead explored a variety of different applications for their canonicalization approach. Examples could include replacing estimated frames with their canonicalized version in existing SoTA 3D equivariant networks or using their frame to replace the one estimated in SHOT.

Overall, the paper lacks compelling applications for the proposed approach so I do not recommend acceptance at this time.

**Questions:**

- Despite the construction of a local Laplacian and the estimation of the Fiedler vector (which are known to be robust under permutations), it appears that the re-ordering of points can regardless be very sensitive to the sampling. For instance, removing and adding a random point to a local neighborhood would probably change the ordering of the points. How is this addressed?

---

> ### Author Response · Authors · 2025-11-18
>
> We thank the reviewer for their insightful comments and for recognizing the value of our canonicalization method.
>
> ---
> ### 1. On the Synergy of Canonicalization and the MLP Application (Weaknesses 1 & 2)
> We appreciate the reviewer's comments on the application of our canonicalization. We would like to clarify that the simple MLP and the curvature pre-training were a **principled design choice**, intended to work in synergy with our canonicalization method.
> * **Standard permutation-invariant models** (e.g., PointNet) use symmetric functions (like max-pooling) that, while ensuring invariance, can discard significant local geometric detail.
> * **Our spectral canonicalization** provides a stable, ordered local frame that is invariant to permutations and rigid transformations **_before_ any learning**. This **built-in invariance** is precisely what enables a 'simple' permutation-variant MLP to focus exclusively on learning the intrinsic geometric properties of the patch, without information loss from pooling.
> By pre-training this tiny operator (0.03M params) to distill a fundamental geometric prior (curvature), we create an efficient building block designed for generalization, avoiding the need for complex, parameter-heavy architectures.
> ---
> ### 2. On Descriptor Retrieval, Generalization, and Efficiency (Weakness 1)
> We thank the reviewer for this point, as it allows us to clarify the most critical context of this experiment: it is a **zero-shot generalization task**.
> 1.  **Zero-Shot Context:** The SOTA methods we are compared against are orders of magnitude larger (e.g., 2.16M to 33.48M parameters) and were trained *specifically* for descriptor matching on the 3DMatch training set. In contrast, our 0.03M parameter model was **never trained on 3DMatch, nor on any real-world scan data**. It was trained *only* on synthetic curvature data.
> 2.  **Generalization as the Goal:** We respectfully position this 65.7% FMR not as an attempt to claim SOTA performance, but as a **strong validation of our model's generalization**. The fact that our tiny, synthetically-trained operator can achieve this result on a complex, unseen, real-world benchmark is, we believe, the central finding. This experiment was designed to be the "investigation of the properties of the proposed MLP as a representation learner" that the reviewer suggested.
> 3.  **Relevance of Efficiency:** We respectfully offer a different perspective on parameter efficiency. For many real-world tasks in robotics, autonomous driving, and AR, efficiency and speed are primary design constraints.
>     * Our lightweight design (0.03M params) translates to a **30x speedup** in descriptor generation over SOTA models like SpinNet. This efficiency stems from two key properties: the operator itself is **extremely lightweight**, and its application is **highly parallelizable** (each descriptor is processed independently).
>     * This combination creates significant **hardware flexibility**:
>         * On **edge devices** with low compute (e.g., CPUs), the **minimal footprint (0.03M)** allows for high-performance 3D perception without requiring a powerful GPU.
>         * On **modern hardware** (e.g., GPUs), its **parallelizable** nature can be fully leveraged for massive throughput and even greater speedups.
>     * We believe achieving strong zero-shot generalization combined with this 30x+ speedup and hardware flexibility makes CanonNet a highly practical and compelling solution.
> ---
> ### 3. On Sensitivity to Point Sampling (Question 1)
> This is an excellent question regarding the method's stability. The robustness of the ordering stems from the fact that it is derived from the Fiedler vector, which provides a spectral embedding $\\phi$ that optimally preserves local structure by minimizing the objective: $$\\sum_{i,j} W_{ij}(\\phi_i - \\phi_j)^2$$ This property is explored in "Laplacian Eigenmaps for Dimensionality Reduction and Data Representation" (Belkin and Niyogi 2003), where $W_{ij}$ is derived from all pairwise distances via the heat kernel.
> Because this embedding depends on the entire local structure (via the fully connected graph), the resulting ordering is **robust to minor changes** in the local point set, such as the addition or removal of a few points. This theoretical stability was also validated empirically in our submission. As shown in **Appendix A (Table 6)**, our method demonstrates **high ordering retention** under significant Gaussian noise, confirming its practical stability.
>
> ---
> ### 4. On Alternative Applications (Weakness 2)
> We agree that utilizing our canonicalization as a plug-in for other frameworks (e.g., equivariant networks, classic descriptors) is a valuable suggestion, and we will incorporate this promising direction into our discussion.
>
> ---
> We hope these clarifications illustrate our contribution: a highly efficient, generalizable geometric operator. We respectfully ask the reviewer to consider these points in their final evaluation.

---

> > ### Comment · Reviewer_dLxm · 2025-11-26
> >
> > Thank you for the refreshing rebuttal.
> >
> > I'm now more confident regarding the stability of the method under perturbations of the neighborhood and am a fan of the frame canonicalization overall.
> >
> > However, the experiments still don't fully convince me about the usefulness of the MLP or pretraining. Adding a Heat Kernel Signature/Wave Kernel Signature baseline to the descriptor experiments would be informative, but I don't think this would get it over the line.
> >
> > More generally, it seems like the frame canonicalization could actually be used to construct a more powerful surface network than just a pointwise MLP. For instance, a convolutional-like model could be developed where the frames are used to create a canonical orientation over a neighborhood, over which either convolutional filters or some kind of local attention could be applied, essentially overcoming the limitations of equivariant surface models which must sacrifice some degree of descriptiveness to preserve equivariance.
> >
> > Unfortunately, I remain unconvinced about the overall approach and will maintain my rating. However, the frame canonicalization is clearly useful, powerful, and interesting. I encourage the authors to find more compelling applications for the canonicalization method, ideally as part of a larger framework for learning on meshes or representation learning.

---

### Official Review · Reviewer_Jf52 · 2025-11-01

**Soundness:** 2
**Presentation:** 2
**Contribution:** 2
**Rating:** 2
**Confidence:** 3

**Summary:**

This paper proposes a local geometry operator for point cloud processing that achieves invariance to rigid transformations and point ordering. The method learns features from a graph constructed using KNN neighborhoods, where the edge weights are computed based on Euclidean distances, ensuring invariance to rotation and translation. The proposed deep learning model is lightweight and shows good performance in curvature estimation and descriptor matching, although it is trained on synthetic data.

**Strengths:**

1. A local feature learning method based on the Laplacian matrix for spectral canonicalization is proposed, which is invariant to permutation and rigid transformations.
2. The model is trained on synthetic data, which reduces the cost of data collection.
3. Experiments are provided for curvature estimation, descriptor matching, and surface classification. The model demonstrates good generalization to unseen data in curvature estimation and descriptor matching.

**Weaknesses:**

1. The main concern lies in the application potential of the proposed local features. Although the features learned from synthetic data show good cross-domain performance, it is unclear whether the model can be extended to large-scale, real-world point cloud understanding tasks such as shape classification (e.g., ShapeNet or Objaverse), part segmentation, or scene-level semantic analysis.
2. An ablation study on varying neighborhood sizes (different K values in KNN) is recommended to evaluate the expressiveness of the learned features.
3. How does the model perform under variations in point cloud density?
4. Given the edge weights defined in Eq. (1), the rigid transformation invariance of the operator is straightforward. Theorem 1 could be moved to the appendix.

**Questions:**

1. Figure 5 is not included in the main paper.
2. For shape or surface classification, since the model is invariant to rotations, the authors could consider comparing it with existing rotation-invariant architectures, such as Vector Neurons [a] and Frame Averaging [b].

[a]. Vector Neurons: A General Framework for SO(3)-Equivariant Networks. ICCV 2021.
[b] Frame Averaging for Invariant and Equivariant Network Design. ICLR 2022.

---

> ### Author Response · Authors · 2025-11-18
>
> We sincerely thank the reviewer for their thoughtful and constructive feedback. We are encouraged that the reviewer recognized the strengths of our spectral canonicalization method, particularly its invariance properties and its ability to generalize well from synthetic data.
>
> We would like to address the reviewer's concerns, focusing on the main point regarding application potential, which we believe we can clarify.
>
> ### 1. On W1 (Application Potential)
>
> > **Reviewer (W1):** "The main concern lies in the application potential... it is unclear whether the model can be extended to large-scale, real-world point cloud understanding tasks such as shape classification... part segmentation, or scene-level semantic analysis."
>
> We thank the reviewer for this crucial point. We would like to clarify that our descriptor matching experiment (Section 4.2) is evaluated on **3DMatch**, which is a standard large-scale, real-world benchmark for point cloud registration.
>
> As shown in Table 2, our model achieves competitive cross-domain performance (65.7% FMR) against much larger methods, despite being extremely lightweight (0.03M params). We believe this directly demonstrates its applicability and robustness in a real-world task.
>
> Our primary contribution is a foundational and highly efficient **local geometry operator**. We intentionally focused on tasks that *directly* validate this geometric fidelity (curvature, descriptors, and surface-type). We agree that scaling this operator to hierarchical tasks like segmentation is excellent future work built upon this foundation.
>
> ### 2. On W2 (Ablation on K) & W3 (Density Variation)
>
> > **Reviewer (W2):** "An ablation study on varying neighborhood sizes (different K values in KNN) is recommended..."
>
> This is a great suggestion. While a full table was omitted for brevity, our development included ablations to find the most efficient neighborhood. We selected **K=20** as it was the smallest neighborhood that yielded robust performance on our geometric tasks (e.g., surface classification), with smaller K values showing degraded performance. We aimed for the smallest K to demonstrate our method's ability to extract rich geometric information from a minimal local signal.
>
> > **Reviewer (W3):** "How does the model perform under variations in point cloud density?"
>
> This is an important point, which we omitted for brevity. Our pipeline is robust to global density variations. We apply a **patch-wise normalization step** where each local patch is rescaled based on a global statistic of the *entire* point cloud (e.g., its bounding box diagonal). This process brings all local patches into a standardized scale for the operator, crucially **preserving the internal local geometry and the relative scale of the patches**, which makes our method robust to global density and scale variations.
>
> ### 3. On Q1 (Missing Figure 5) & W4 (Theorem 1)
>
> > **Reviewer (Q1):** "Figure 5 is not included in the main paper."
>
> We apologize for the confusion. Figure 5, which shows the surface classification results under noise (Section 4.3), is located in the **Appendix**. The figure shows accuracy degrading gracefully from 98% (0% noise) to 80% (10% noise), supporting our robustness claims.
>
> > **Reviewer (W4):** "Theorem 1 could be moved to the appendix."
>
> We thank the reviewer for this suggestion to improve flow. We will move Theorem 1 to the appendix in the final version.
>
> ### 4. On Q2 (Suggested Comparison to VN/FA)
>
> > **Reviewer (Q2):** "...authors could consider comparing it with existing rotation-invariant architectures, such as Vector Neurons [a] and Frame Averaging [b]."
>
> We thank the reviewer for suggesting these relevant works. A direct comparison is less straightforward as our methods address invariance differently. VN/FA introduce *equivariance* into the network backbone, which still requires a base permutation-invariant model.
>
> In contrast, our method is a **preprocessing pipeline** that establishes a canonical representation, making the input *invariant* to both permutation and rigid transforms *before* learning. This allows us to use a simple, highly-efficient MLP, which is a different architectural paradigm. We will add this clarification to our related work.
>
> ---
> We hope these clarifications address the reviewer's concerns and highlight our contribution as a highly efficient and robust foundational operator for real-world geometric tasks. We thank the reviewer again for their time and valuable feedback.

---

> > ### Comment · Reviewer_Jf52 · 2025-11-28
> >
> > Thank the authors for the rebuttal and the clarification regarding canonicalization. However, using a preprocessing step for canonicalization weakens the technical contribution, as many existing 3D neural networks could similarly benefit from pre-processed input features. Given the focus on learning invariant features, including comparisons with equivariant networks would be more appropriate.

---

### Meta-Review · Area_Chair_M8a8 · 2025-12-03

**Summary:**

The reviewers find the core idea of learning canonical, curvature-aware local features for point clouds interesting, but they raise several major concerns:

1. Limited motivation and justification. The design of the canonical local frame and its ordering/orientation is not well justified or analyzed. It remains unclear why curvature estimation in the proposed synthetic form is important for representation learning, and how broadly useful the resulting features are.

2. Weak and narrow experimental validation. The experiments focus on synthetic curvature estimation and descriptor retrieval on limited synthetic datasets. The method is not evaluated on standard real-world 3D tasks, and some key factors are not systematically studied. On descriptor retrieval, the approach is not competitive with existing methods, and the claimed parameter efficiency is not sufficient to compensate for the performance gap.

3. Missing broader applications and analysis. Reviewers feel that the frame canonicalization component is potentially more generally useful, but the paper does not explore such applications or ablations. This limits the demonstrated impact and generality of the proposed idea.

4. Presentation and clarity issues. The paper would benefit from clearer explanations and stronger organization; some theoretical parts could be adjusted, and the overall writing needs substantial improvement.

Two out of three reviewers acknowledged the rebuttal, but decided to maintain their original scores due to unaddressed concerns. Overall, the AC recommends rejecting the paper in its current form.

**Reviewer Concerns:**

Reviewer Jf52’s central concerns remain: using canonicalization as a preprocessing step weakens the claimed contribution (since many 3D networks could benefit similarly), and the work still does not position itself convincingly relative to equivariant/invariant architectures.

Reviewer dLxm maintained the score, since the usefulness of the specific MLP architecture and pretraining strategy is not convincingly demonstrated; the experimental evidence remains weak compared to existing descriptor-learning methods; and the paper does not explore more powerful or diverse applications of canonicalization.

There is no follow-up response from Reviewer 8aXT. The AC has carefully read the rebuttal but finds that the main concerns raised in the original review are not adequately resolved.

**Reviewer Scores:**

Reviewer Jf52: 2 (the reviewed acknowledged the rebuttal but maintained the score)
Reviewer dLxm: 4 (the reviewed acknowledged the rebuttal but maintained the score)
Reviewer 8aXT: 4 (no follow-up)

---

### Decision · Program_Chairs · 2026-01-26

Reject